# AutoTransfer: AutoML with Knowledge Transfer - An Application to Graph Neural Networks

**Kaidi Cao**   **Jiaxuan You**   **Jiaju Liu**   **Jure Leskovec**

Department of Computer Science, Stanford University

`{kaidicao, jiaxuan, jiajuliu, jure}@cs.stanford.edu`

## Abstract

AutoML has demonstrated remarkable success in finding an effective neural architecture for a given machine learning task defined by a specific dataset and an evaluation metric. However, most present AutoML techniques consider each task independently from scratch, which requires exploring many architectures, leading to high computational costs. Here we propose AutoTransfer, an AutoML solution that improves search efficiency by transferring the prior architectural design knowledge to the novel task of interest. Our key innovation includes a *task-model bank* that captures the model performance over a diverse set of GNN architectures and tasks, and a computationally efficient *task embedding* that can accurately measure the similarity among different tasks. Based on the task-model bank and the task embeddings, we estimate the design priors of desirable models of the novel task, by aggregating a similarity-weighted sum of the top-K design distributions on tasks that are similar to the task of interest. The computed design priors can be used with any AutoML search algorithm. We evaluate AutoTransfer on six datasets in the graph machine learning domain. Experiments demonstrate that (i) our proposed task embedding can be computed efficiently, and that tasks with similar embeddings have similar best-performing architectures; (ii) AutoTransfer significantly improves search efficiency with the transferred design priors, reducing the number of explored architectures by *an order of magnitude*. Finally, we release GNN-Bank-101, a large-scale dataset of detailed GNN training information of 120,000 task-model combinations to facilitate and inspire future research.

## 1 Introduction

Deep neural networks are highly modular, requiring many design decisions to be made regarding network architecture and hyperparameters. These design decisions form a search space that is nonconvex and costly even for experts to optimize over, especially when the optimization must be repeated from scratch for each new use case. Automated machine learning (AutoML) is an active research area that aims to reduce the human effort required for architecture design that usually covers hyperparameter optimization and neural architecture search. AutoML has demonstrated success (Zoph and Le, 2016; Pham et al., 2018; Zoph et al., 2018; Cai et al., 2018; He et al., 2018; Guo et al., 2020; Erickson et al., 2020; LeDell and Poirier, 2020) in many application domains.

Finding a reasonably good model for a new learning *task*[1] in a computationally efficient manner is crucial for making deep learning accessible to domain experts with diverse backgrounds. Efficient AutoML is especially important in domains where the best architectures/hyperparameters are highly sensitive to the task. A notable example is the domain of graph learning[2]. *First*, graph learning methods receive input data composed of *a variety of data types* and optimize over tasks that span an equally *diverse set of domains and modalities* such as recommendation (Ying et al., 2018; He et al., 2020), physical simulation (Sanchez-Gonzalez et al., 2020; Pfaff et al., 2020), and bioinformatics (Zitnik et al., 2018). This differs from computer vision and natural language processing where the

---

[1]In this paper, we refer to a *task* as a given dataset with an evaluation metric/loss, *e.g.*, cross-entropy loss on node classification on the Cora dataset.

[2]We focus on the graph learning domain in this paper. AutoTransfer can be generalized to other domains.

input data has a predefined, fixed structure that can be shared across different neural architectures. *Second*, neural networks that operate on graphs come with *a rich set of design choices* and *a large set of parameters to explore*. However, unlike other domains where a few pre-trained architectures such as ResNet (He et al., 2016) and GPT-3 (Brown et al., 2020) dominate the benchmarks, it has been shown that the best graph neural network (GNN) design is highly task-dependent (You et al., 2020).

Although AutoML as a research domain is evolving fast, existing AutoML solutions have massive computational overhead when finding a good model for a *new* learning task is the goal. Most present AutoML techniques consider each task independently and in isolation, therefore they require redoing the search from scratch for each new task. This approach ignores the potentially valuable architectural design knowledge obtained from the previous tasks, and inevitably leads to a high computational cost. The issue is especially significant in the graph learning domain Gao et al. (2019); Zhou et al. (2019), due to the challenges of diverse task types and the huge design space that are discussed above.

Here we propose AUTOTRANSFER[3], an AutoML solution that drastically improves AutoML architecture search by transferring previous architectural design knowledge to the task of interest. Our key innovation is to introduce a *task-model bank* that stores the performance of a diverse set of GNN architectures and tasks to guide the search algorithm. To enable knowledge transfer, we define a *task embedding* space such that tasks close in the embedding space have similar corresponding top-performing architectures. The challenge here is that the task embedding needs to capture the performance rankings of different architectures on different datasets, while being efficient to compute. Our innovation here is to embed a task by using the condition number of its Fisher Information Matrix of various *randomly initialized* models and also a learning scheme with an empirical generalization guarantee. This way we implicitly capture the properties of the learning task, while being orders of magnitudes faster (within seconds). We then estimate the design prior of desirable models for the new task, by aggregating design distributions on tasks that are close to the task of interest. Finally, we initiate a hyperparameter search algorithm with the task-informed design prior computed.

We evaluate AUTOTRANSFER on six datasets, including both node classification and graph classification tasks. We show that our proposed task embeddings can be computed efficiently and the distance measured between tasks correlates highly (0.43 Kendall correlation) with model performance rankings. Furthermore, we present AUTOTRANSFER significantly improves search efficiency when using the transferred design prior. AUTOTRANSFER reduces the number of explored architectures needed to reach a target accuracy by *an order of magnitude* compared to SOTA. Finally, we release GNN-BANK-101—the first large-scale database containing detailed performance records for 120,000 task-model combinations which were trained with 16,128 GPU hours—to facilitate future research.

## 2   RELATED WORK

In this section, we summarize the related work on AutoML regarding its applications on GNNs, the common search algorithms, and pioneering work regarding transfer learning and task embeddings.

**AutoML for GNNs.** Neural architecture search (NAS), a unique and popular form of AutoML for deep learning, can be divided into two categories: multi-trial NAS and one-shot NAS. During multi-trial NAS, each sampled architecture is trained separately. GraphNAS (Gao et al., 2020) and Auto-GNN (Zhou et al., 2019) are typical multi-trial NAS algorithms on GNNs which adopt an RNN controller that learns to suggest better sets of configurations through reinforcement learning. One-shot NAS (*e.g.*, (Liu et al., 2018; Qin et al., 2021; Li et al., 2021)) involves encapsulating the entire model space in one super-model, training the super-model once, and then iteratively sampling sub-models from the super-model to find the best one. In addition, there is work that explicitly studies fine-grained design choices such as data augmentation (You et al., 2021), message passing layer type (Cai et al., 2021; Ding et al., 2021; Zhao et al., 2021), and graph pooling (Wei et al., 2021). Notably, AUTOTRANSFER is the *first* AutoML solution for GNNs that efficiently transfer design knowledge across tasks.

**HPO Algorithms.** Hyperparameter Optimization (HPO) algorithms search for the optimal model hyperparameters by iteratively suggesting a set of hyperparameters and evaluating their performance. Random search samples hyperparameters from the search space with equal probability. Despite not

---

[3]Source code is available at `https://github.com/snap-stanford/AutoTransfer`.

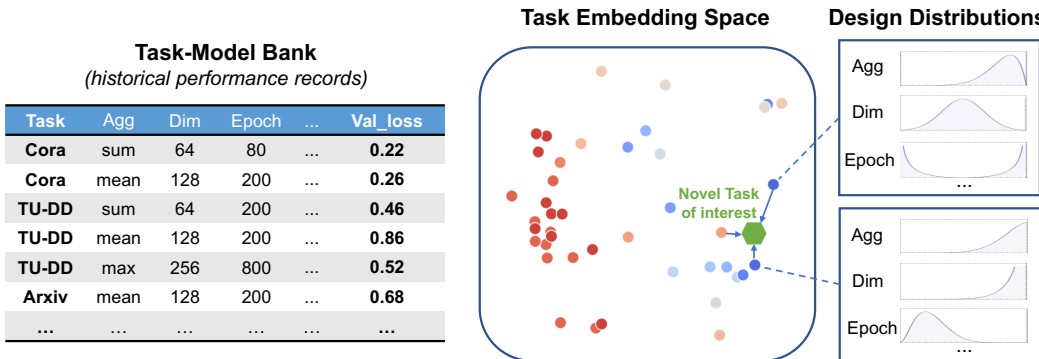

Figure 1: **Overview of AUTOTRANSFER**. **Left**: We introduce GNN-BANK-101, a large database containing a diverse set of GNN architectures and hyperparameters applied to different tasks, along with their training/evaluation statistics. **Middle**: We introduce a *task embedding space*, where each point corresponds to a different task. Tasks close in the embedding space have similar corresponding top-performing models. **Right**: Given a new task of interest, we guide the AutoML search by referencing the *design distributions* of the most similar tasks in the task embedding space.

learning from previous trials, random search is commonly used for its simplicity and is much more efficient than grid search (Bergstra and Bengio, 2012). The TPE algorithm (Bergstra et al., 2011) builds a probabilistic model of task performance over the hyperparameter space and uses the results of past trials to choose the most promising next configuration to train, which the TPE algorithm defines as maximizing the Expected Improvement value (Jones, 2001). Evolutionary algorithms (Real et al., 2017; Jaderberg et al., 2017) train multiple models in parallel and replace poorly performing models with "mutated" copies of the current best models. AUTOTRANSFER is a general AutoML solution and can be applied in combination with any of these HPO algorithms.

**Transfer Learning in AutoML.** Wong et al. (2018) proposed to transfer knowledge across tasks by reloading the controller of reinforcement learning search algorithms. However, this method assumes that the search space on different tasks starts with the same learned prior. Unlike AUTOTRANSFER, it cannot address the core challenge in GNN AutoML: the best GNN design is highly task-specific. GraphGym (You et al., 2020) attempts to transfer the best architecture design directly with a metric space that measures task similarity. GraphGym (You et al., 2020) computes task similarity by training a set of 12 "anchor models" to convergence which is computationally expensive. In contrast, AUTOTRANSFER designs light-weight task embeddings requiring minimal computations overhead. Additionally, Zhao and Bilen (2021); Li et al. (2021) proposes to conduct architecture search on a proxy subset of the whole dataset and later transfer the best searched architecture on the full dataset. Jeong et al. (2021) studies a similar setting in vision domain.

**Task Embedding.** There is prior research trying to quantify task embeddings and similarities. Similar to GraphGym, Taskonomy (Zamir et al., 2018) estimates the task affinity matrix by summarizing final losses/evaluation metrics using an Analytic Hierarchy Process (Saaty, 1987). From a different perspective, Task2Vec (Achille et al., 2019) generates task embeddings for a given task using the Fisher Information Matrix associated with a pre-trained probe network. This probe network is shared across tasks and allows Task2Vec to estimate the Fisher Information Matrix of different image datasets. Le et al. (2022) extends a similar idea to neural architecture search. The aforementioned task embeddings cannot be directly applied to GNNs as the inputs do not align across datasets. AUTOTRANSFER avoids the bottleneck by using asymptotic statistics of the Fisher Information Matrix with randomly initiated weights.

## 3 PROBLEM FORMULATION AND PRELIMINARIES

We first introduce formal definitions of data structures relevant to AUTOTRANSFER.

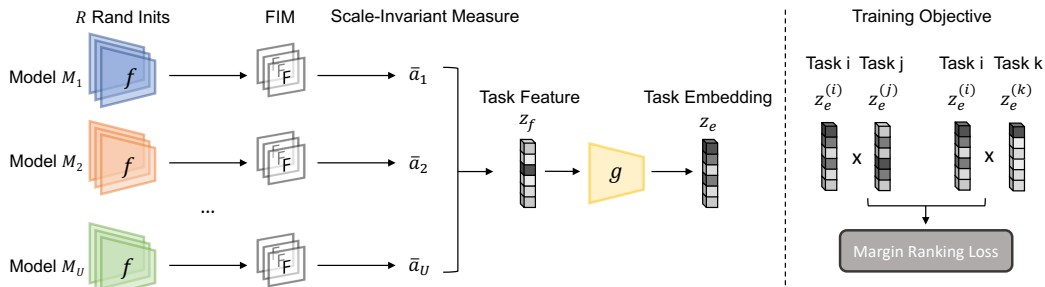

Figure 2: **Pipeline for extracting task embeddings**. **Left**: To efficiently embed a task, we first extract task features by concatenating features measured from $R$ randomly initialized anchor models. Then, we introduce a projection function $g(\cdot)$ with learned weights to transform the task features into task embeddings. **Right**: Training objective for optimizing $g(\cdot)$ with triplet supervision.

**Definition 1 (Task)** *We denote a task as $T = (\mathcal{D}, \mathcal{L}(\cdot))$, consisting of a dataset $\mathcal{D}$ and a loss function $\mathcal{L}(\cdot)$ related to the evaluation metric.*

For each training attempt on a task $T^{(i)}$, we can record its model architecture $M_j$, hyperparameters $H_j$, and corresponding value of loss $l_j$, *i.e.*, $(M_j, H_j, l_j)$. We propose to maintain a task-model bank to facilitate knowledge transfer to future novel tasks.

**Definition 2 (Task-Model Bank)** *A task-model bank $\mathcal{B}$ is defined as a collection of tasks, each with multiple training attempts, in the form of $\mathcal{B} = \{(T^{(i)}, \{(M_j^{(i)}, H_j^{(i)}, l_j^{(i)})\})\}$.*

**AutoML with Knowledge Transfer.** Suppose we have a task-model bank $\mathcal{B}$. Given a novel task $T^{(n)}$ which has not been seen before, our goal is to quickly find a model that works reasonably well on the novel task by utilizing knowledge from the task-model bank.

In this paper, we focus on AutoML for graph learning tasks, though our developed technique is general and can be applied to other domains. We define the input graph as $G = \{V, E\}$, where $V$ is the node set and $E \subseteq V \times V$ is the edge set. Furthermore, let $y$ denote its output labels, which can be node-level, edge-level or graph-level. A GNN parameterized by weights $\theta$ outputs a posterior distribution $\mathcal{P}(G, y, \theta)$ for label predictions.

## 4 PROPOSED SOLUTION: AUTOTRANSFER

In this section, we introduce the proposed AUTOTRANSFER solution. AUTOTRANSFER uses the *task embedding space* as a tool to understand the relevance of previous architectural designs to the target task. The designed task embedding captures the performance rankings of different architectures on different tasks while also being efficient to compute. We first introduce a theoretically motivated solution to extract a scale-invariant performance representation of each task-model pair. We use these representations to construct task features and further learn task embeddings. These embeddings form the task embedding space that we finally use during the AutoML search.

### 4.1 BASICS OF THE FISHER INFORMATION MATRIX (FIM)

Given a GNN defined above, its Fisher Information Matrix (FIM) $F$ is defined as

$$F = \mathbb{E}_{G,y}[\nabla_\theta \log \mathcal{P}(G, y, \theta) \, \nabla_\theta \log \mathcal{P}(G, y, \theta)^\top].$$

which formally is the expected covariance of the scores with respect to the model parameters. There are two popular geometric views for the FIM. First, the FIM is an upper bound of the Hessian and coincides with the Hessian if the gradient is 0. Thus, the FIM characterizes the local landscape of the loss function near the global minimum. Second, similar to the Hessian, the FIM models the loss

landscape with respect not to the input space, but to the parameter space. In the information geometry view, if we add a small perturbation to the parameter space, we have

$$\mathrm{KL}(\mathcal{P}(G, y, \theta) \| \mathcal{P}(G, y, \theta + d\theta)) = d\theta^\top F d\theta.$$

where $\mathrm{KL}(\cdot, \cdot)$ stands for Kullback–Leibler divergence. It means that the parameter space of a model forms a Riemannian manifold and the FIM works as its Riemannian metric. The FIM thus allows us to quantify the importance of a model's weights in a way that is applicable to different architectures.

## 4.2 FIM-based Task Features

**Scale-invariant Representation of Task-Model Pairs.** We aim to find a scale-invariant representation for each task-model pair which will form the basis for constructing task features. The major challenge in using the FIM to represent GNN performance is that graph datasets do not have a universal, fixed input structure, so it is infeasible to find a single pre-trained model and extract its FIM. However, training multiple networks poses a problem as the FIMs computed for different networks are not directly comparable. We choose to use multiple networks but additionally propose to use asymptotic statistics of the FIM associated with randomly initialized weights. The theoretical justification for the relationship between the asymptotic statistics of the FIM and the trainability of neural networks was studied in (Karakida et al., 2019; Pennington and Worah, 2018) to which we refer the readers. We hypothesize that such a measure of trainability encodes loss landscapes and generalization ability and thus correlates with final model performance on the task. Another issue that relates to input structures of graph datasets is that different models have different number of parameters. Despite some specially designed architectures, *e.g.*, (Lee et al., 2019; Ma et al., 2019), most GNN architecture design can be represented as a sequence of pre-processing layers, message passing layers, and post-processing layers. Pre-process layers and post-process layers are Multilayer Perceptron (MLP) layers, of which the dimensions vary across different tasks due to different input/output structures. Message passing layers are commonly regarded as the key design for GNNs and the number of weight parameters can remain the same across tasks. In this light, we only consider the FIM with respect to the parameters in message passing layers so that the number of parameters considered stays the same for all datasets. We note that such formulation has its limitations, in the sense that it cannot cover all the GNN designs in the literature. We leave potential extensions with better coverage for future work. We further approximate the FIM by only considering the diagonal entries, which implicitly neglects the correlations between parameters. We note that this is common practice when analyzing the FIMs of deep neural networks, as the full FIM is massive (quadratic in the number of parameters) and infeasible to compute even on modern hardware. Similar to Pennington and Worah (2018), we consider the first two moments of FIM

$$m_1 = \frac{1}{n} \mathrm{tr}[F] \quad \text{and} \quad m_2 = \frac{1}{n} \mathrm{tr}[F^2] \tag{1}$$

and use $\alpha = m_2/m_1^2$ as the scale-invariant representation. The computed $\alpha$ is lower bounded by 1 and captures how concentrated the spectrum is. A small $\alpha$ indicates the loss landscape is flat, and its corresponding model design enjoys fast first-order optimization and potentially better generalization. To encode label space information into each task, we propose to train only the last linear layer of each model on a given task, which can be done efficiently. The parameters in other layers are frozen after being randomly initialized. We take the average over $R$ initializations to estimate the average $\bar{\alpha}$.

**Constructing Task Features.** We denote task features as measures extracted from each task that characterize its important traits. The design of task features should reflect our final objective: to use these features to identify similar tasks and transfer the best design distributions. Thus, we select $U$ model designs as anchor models and concatenate the scale-invariant representations $\bar{a}_u$ of each design as task features. To retain only the relative ranking among anchor model designs, we normalize the concatenated feature vector to a scale of 1. We let $\mathbf{z}_f$ denote the normalized task feature.

## 4.3 From Task Features to Task Embeddings

The task feature $\mathbf{z}_f$ introduced above can be regarded as a means of feature engineering. We construct the feature vector with domain knowledge, but there is no guarantee it functions as anticipated. We thus propose to learn a projection function $g(\cdot) : \mathbb{R}^U \to \mathbb{R}^D$ that maps task feature $\mathbf{z}_f$ to final task

embedding $z_e = g(z_f)$. We do not have any pointwise supervision that can be used as the training objective. Instead, we consider the metric space defined by GraphGym. The distance function in GraphGym - computed using the Kendall rank correlation between performance rankings of anchor models trained on the two compared tasks - correlates nicely with our desired knowledge transfer objective. It is not meaningful to enforce that task embeddings mimic GraphGym's exact metric space, as GraphGym's metric space can still contain noise, or does not fully align with the transfer objective. We consider a surrogate loss that enforces only the rank order among tasks. To illustrate, let us consider tasks $T^{(i)}, T^{(j)}, T^{(k)}$ and their corresponding task embeddings, $z_e^{(i)}, z_e^{(j)}, z_e^{(k)}$. Note that $z_e$ is normalized to 1 so $z_e^{(i)^\top} z_e^{(j)}$ measures the cosine similarity between tasks $T^{(i)}$ and $T^{(j)}$. Let $d_g(\cdot, \cdot)$ denote the distance estimated by GraphGym. We want to enforce

$$z_e^{(i)^\top} z_e^{(j)} > z_e^{(i)^\top} z_e^{(k)} \quad \text{if} \quad d_g(T^{(i)}, T^{(j)}) < d_g(T^{(i)}, T^{(k)}).$$

To achieve this, we use the margin ranking loss as our surrogate supervised objective function:

$$\mathcal{L}_r(z_e^{(i)}, z_e^{(j)}, z_e^{(k)}, y) = \max(0, -y \cdot (z_e^{(i)^\top} z_e^{(j)} - z_e^{(i)^\top} z_e^{(k)}) + \text{margin}). \tag{2}$$

Here if $d_g(T^{(i)}, T^{(j)}) < d_g(T^{(i)}, T^{(k)})$, then we have its corresponding label $y = 1$, and $y = -1$ otherwise. Our final task embedding space is then a FIM-based metric space with cosine distance function, where the distance is defined as $d_e(T^{(i)}, T^{(j)}) = 1 - z_e^{(i)^\top} z_e^{(j)}$. Please refer to the detailed training pipeline at Algorithm 2 in the Appendix.

## 4.4 AutoML Search Algorithm with Task Embeddings

To transfer knowledge to a novel task, a naïve idea would be to directly carry over the best model configuration from the closest task in the bank. However, even a high Kendall rank correlation between model performance rankings of two tasks $T^{(i)}, T^{(j)}$ does not guarantee the best model configuration in task $T^{(i)}$ will also achieve the best performance on task $T^{(j)}$. In addition, since task similarities are subject to noise, this naïve solution may struggle when there exist multiple reference tasks that are all highly similar.

To make the knowledge transfer more robust to such failure cases, we introduce the notion of design distributions that depend on top performing model designs and propose to transfer design distributions rather than the best design configurations. Formally, consider a task $T^{(i)}$ in the task-model bank $\mathcal{B}$, associated with its trials $\{(M_j^{(i)}, H_j^{(i)}, l_j^{(i)})\}$. We can summarize its designs as a list of configurations $C = \{c_1, \ldots, c_W\}$, such that all potential combinations of model architectures $M$ and hyperparameters $H$ fall under the Cartesian product of the configurations. For example, $c_1$ could be the instantiation of aggregation layers, and $c_2$ could be the start learning rate. We then define design distributions as random variables $\mathsf{c}_1, \mathsf{c}_2, \ldots, \mathsf{c}_W$, each corresponding to a hyperparameter. Each random variable $\mathsf{c}$ is defined as the frequency distribution of the design choices used in the top $K$ trials. We multiply all distributions for the individual configurations $\{\mathsf{c}_1, \ldots, \mathsf{c}_W\}$ to approximate the overall task's design distribution $\mathcal{P}(C|T^{(i)}) = \prod_w \mathcal{P}(\mathsf{c}_w|T^{(i)})$.

During inference, given a novel task $T^{(n)}$, we select a close task subset $\mathcal{S}$ by thresholding task embedding distances, i.e., $\mathcal{S} = \{T^{(i)}|d_e(T^{(n)}, T^{(i)}) \leq d_{\text{thres}}\}$. We then derive the transferred design prior $\mathcal{P}_t(C|T^{(n)})$ of the novel task by weighting design distributions from the close task subset $\mathcal{S}$.

$$\mathcal{P}_t(C|T^{(n)}) = \frac{\sum_{T^{(i)} \in \mathcal{S}} \frac{1}{d_e(T^{(n)}, T^{(i)})} \mathcal{P}(C|T^{(i)})}{\sum_{T^{(i)} \in \mathcal{S}} \frac{1}{d_e(T^{(n)}, T^{(i)})}}. \tag{3}$$

The inferred design prior for the novel task can then be used to guide various search algorithms. The most natural choice for a few trial regime is random search. Rather than sampling each design configuration following a uniform distribution, we propose to sample from the task-informed design prior $\mathcal{P}_t(C|T^{(n)})$. Please refer to Appendix A to check how we augment other search algorithms.

For AutoTransfer, we can preprocess the task-model bank $\mathcal{B}$ into $\mathcal{B}_p = \{(\mathcal{D}^{(i)}, \mathcal{L}^{(i)}(\cdot)), z_e^{(i)}, \mathcal{P}(C|T^{(i)})\}$ as our pipeline only requires using task embedding $z_e^{(i)}$ and design distribution $\mathcal{P}(C|T^{(i)})$ rather than detailed training trials. A detailed search pipeline is summarized in Algorithm 1.

---

**Algorithm 1** Summary of AUTOTRANSFER search pipeline

---

**Require:** A processed task-model bank $\mathcal{B}_p = \{(\mathcal{D}^{(i)}, \mathcal{L}^{(i)}(\cdot)), \boldsymbol{z}_e^{(i)}, \mathcal{P}(C|T^{(i)})\}$, a novel task $T^{(n)} = \left(\mathcal{D}^{(n)}, \mathcal{L}^{(n)}(\cdot)\right)$, $U$ anchor models $M_1, ..., M_U$, $R$ specifies the number of repeats.
  1: **for** $u = 1$ to $U$ **do**
  2:    **for** $r = 1$ to $R$ **do**
  3:       Initialize weights $\theta$ for anchor model $M_u$ randomly
  4:       Estimate FIM $F \leftarrow \mathbb{E}_{\mathcal{D}}[\nabla_\theta \log \mathcal{P}(M_u, y, \theta) \nabla_\theta \log \mathcal{P}(M_u, y, \theta)^\top]$
  5:       Extract scale-invariant representation $a_u^{(v)} \leftarrow m_2/m_1^2$ following Eq. 1
  6:    **end for**
  7:    $\bar{a}_u \leftarrow \text{mean}(a_u^{(1)}, a_u^{(2)}, ..., a_u^{(V)})$
  8: **end for**
  9: $\boldsymbol{z}_f^{(n)} \leftarrow \text{concat}(\bar{a}_1, \bar{a}_2, ..., \bar{a}_U)$
 10: $\boldsymbol{z}_e^{(n)} \leftarrow g(\boldsymbol{z}_f^{(n)})$
 11: Select close task subset $\mathcal{S} \leftarrow \{T^{(i)}|1 - \boldsymbol{z}_e^{(n)^\top} \boldsymbol{z}_e^{(i)} \leq d_{\text{thres}}\}$
 12: Get design prior $\mathcal{P}_t(C|T^{(n)})$ by aggregating subset $\mathcal{S}$ following Eq. 3
 13: Start a HPO search algorithm with the task-informed design prior $\mathcal{P}_t(C|T^{(n)})$

---

## 5 EXPERIMENTS

### 5.1 EXPERIMENTAL SETUP

**Task-Model Bank: GNN-BANK-101.**

To facilitate AutoML research with knowledge transfer, we collected GNN-BANK-101 as the first large-scale graph database that records reproducible design configurations and detailed training performance on a variety of tasks. Specifically, GNN-BANK-101 currently includes six tasks for node classification (AmazonComputers (Shchur et al., 2018), AmazonPhoto (Shchur et al., 2018), CiteSeer (Yang et al., 2016), CoauthorCS (Shchur et al., 2018), CoauthorPhysics (Shchur et al., 2018), Cora (Yang et al., 2016)) and six tasks for graph classification (PROTEINS (Ivanov et al., 2019), BZR (Ivanov et al., 2019), COX2 (Ivanov et al., 2019), DD (Ivanov et al., 2019), ENZYMES (Ivanov et al., 2019), IMDB-M (Ivanov et al., 2019)). Our design space follows (You et al., 2020), and we extend the design space to include various commonly adopted graph convolution and activation layers. We extensively run 10,000 different models for each task, leading to 120,000 total task-model combinations, and record all training information including train/val/test loss.

**Benchmark Datasets.** We benchmark AUTOTRANSFER on six different datasets following prior work (Qin et al., 2021). Our datasets include three standard node classification datasets (Coauthor-Physics (Shchur et al., 2018), CoraFull (Bojchevski and Günnemann, 2017) and OGB-Arxiv (Hu et al., 2020)), as well as three standard benchmark graph classification datasets, (COX2 (Ivanov et al., 2019), IMDB-M (Ivanov et al., 2019) and PROTEINS (Ivanov et al., 2019)). CoauthorPhysics and CoraFull are transductive node classification datasets, so we randomly assign nodes into train/valid/test sets following a 50%:25%:25% split (Qin et al., 2021). We randomly split graphs following a 80%:10%:10% split for the three graph classification datasets (Qin et al., 2021). We follow the default train/valid/test split for the OGB-Arxiv dataset (Hu et al., 2020). To make sure there is no information leakage, we temporarily *remove* all records related to the task from our task-model bank if the dataset we benchmark was collected in the task-model bank.

**Baselines.** We compare our methods with the state-of-the-art approaches for GNN AutoML. We use GCN and GAT with default architectures following their original implementation as baselines. For multi-trial NAS methods, we consider GraphNAS (Gao et al., 2020). For one-shot NAS methods, we include DARTS (Liu et al., 2018) and GASSO (Qin et al., 2021). GASSO is designed for transductive settings, so we omit it for graph classification benchmarks. We further provide results of HPO algorithms based on our proposed search space as baselines: Random, Evolution, TPE (Bergstra et al., 2011) and HyperBand (Li et al., 2017).

We by default allow searching 30 trials maximum for all the algorithms, *i.e.*, an algorithm can train 30 different models and collect the model with the best accuracy. We use the default setting for one-shot

Table 1: Performance comparisons of AUTOTRANSFER and other baselines. We report the average test accuracy and the standard deviation over ten runs. With only 3 trials AUTOTRANSFER already outperform most SOTA baselines with 30 trials.

| Method | Physics | Node CoraFull | OGB-Arxiv | COX2 | Graph IMDB | PROTEINS |
|---|---|---|---|---|---|---|
| GCN (30 trials) | 95.88±0.16 | 67.12±0.52 | 70.46±0.18 | 79.23±2.19 | 50.40±3.02 | 74.84±2.82 |
| GAT (30 trials) | 95.71±0.24 | 65.92±0.68 | 68.82±0.32 | 81.56±4.17 | 49.67±4.30 | 75.30±3.72 |
| GraphNAS (30 trials) | 92.77±0.84 | 63.13±3.28 | 65.90±2.64 | 77.73±1.40 | 46.93±3.94 | 72.51±3.36 |
| DARTS | 95.28±1.67 | 67.59±2.85 | 69.02±1.18 | 79.82±3.15 | 50.26±4.08 | 75.04±3.81 |
| GASSO[4] | 96.38 | 68.89 | 70.52 | - | - | - |
| Random (3 trials) | 95.16±0.55 | 61.24±4.04 | 67.92±1.92 | 76.88±3.17 | 45.79±4.39 | 72.47±2.57 |
| TPE (30 trials) | 96.41±0.36 | 66.37±1.73 | 71.35±0.44 | 82.27±2.01 | 50.33±4.00 | 79.46±1.28 |
| HyperBand (30 trials) | 96.56±0.30 | 67.75±1.24 | 71.60±0.36 | 82.21±1.79 | 50.86±3.45 | 79.32±1.16 |
| **AUTOTRANSFER (3 trials)** | 96.64±0.42 | 69.27±0.76 | 71.42±0.39 | 82.13±1.59 | 52.33±2.13 | 77.81±2.19 |
| **AUTOTRANSFER (30 trials)** | 96.91±0.27 | 70.05±0.42 | 72.21±0.27 | 86.52±1.58 | 54.93±1.23 | 81.25±1.17 |

NAS algorithms (DARTS and GASSO), as they only train a super-model once and can efficiently evaluate different architectures. We are mostly interested in studying the few-trial regime where most advanced search algorithms degrade to random search. Thus we additionally include a random search (3 trials) baseline where we pick the best model out of only 3 trials.

## 5.2 EXPERIMENTS ON SEARCH EFFICIENCY

We evaluate AUTOTRANSFER by reporting the average best test accuracy among all trials considered over ten runs of each algorithm in Table 1. The test accuracy collected for each trial is selected at the epoch with the best validation accuracy. By comparing results from random search (3 trials) and AUTOTRANSFER (3 trials), we show that our transferred task-informed design prior significantly improves test accuracy in the few-trial regime, and can be very useful in environments that are where computationally constrained. Even if we increase the number of search trials to 30, AUTOTRANSFER still demonstrates non-trivial improvement compared with TPE, indicating that our proposed pipeline has advantages even when computational resources are abundant. Notably, with only 3 search trials, AUTOTRANSFER surpasses most of the baselines, even those that use 30 trials.

To better understand the sample efficiency of AUTOTRANSFER, we plot the best test accuracy found at each trial in Figure 3 for OGB-Arxiv and TU-PROTEINS datasets. We notice that the advanced search algorithms (Evolution and TPE) have no advantages over random search at the few-trial regime since the amount of prior search data is not yet sufficient to infer potentially better design configurations. On the contrary, by sampling from the transferred design prior, AUTOTRANSFER achieves significantly better average test accuracy in the first few trials. The best test accuracy at trial 3 of AUTOTRANSFER surpasses its counterpart at trial 10 for every other method.

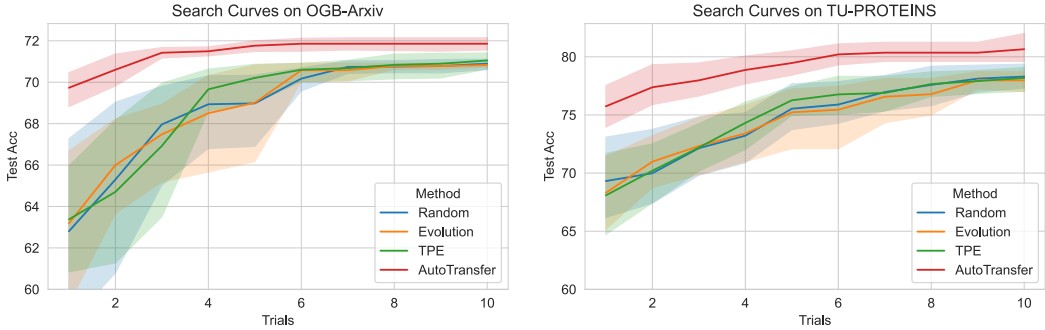

Figure 3: Performance comparisons in the few-trial regime. At trial $t$, we plot the best test accuracy among all models searched from trial 1 to trial $t$. **AUTOTRANSFER can reduce the number of trials needed to search by an order of magnitude** (see also Table 4 in Appendix).

## 5.3 ANALYSIS OF TASK EMBEDDINGS

**Qualitative analysis of task features.** To examine the quality of the proposed task features, we visualize the proposed task similarity matrix (Figure 4 (b)) along with the task similarity matrix (Figure 4 (a)) proposed in GraphGym. We show that our proposed task similarity matrix captures similar patterns as GraphGym's task similarity matrix while being computed much more efficiently

---

[2]Results come from the original paper (Qin et al., 2021).

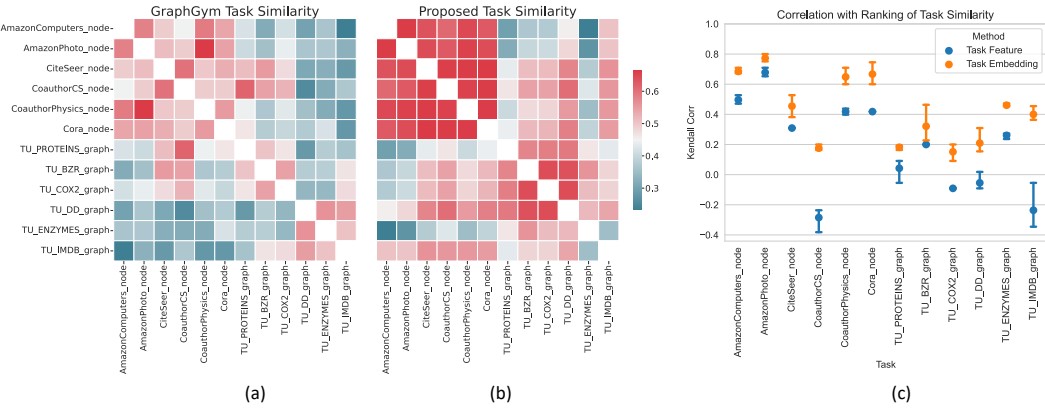

Figure 4: (a) GraphGym's task similarity between all pairs of tasks (computed from the Kendall rank correlation between performance rankings of models trained on the two compared tasks), a higher value represents a higher similarity. (b) The proposed task similarity computed by computing the dot-product between extracted task features. (c) The Kendall rank correlation of similarity rankings of the other tasks with respect to the central task between the proposed method and GraphGym.

by omitting training. We notice that the same type of tasks, *i.e.*, node classification and graph classification, share more similarities within each group. As a sanity check, we examined that the closest task in the bank with respect to CoraFull is Cora. The top 3 closest tasks for OGB-Arxiv are AmazonComputers, AmazonPhoto, and CoauthorPhysics, all of which are node classification tasks.

**Generalization of projection function** $g(\cdot)$. To show the proposed projection function $g(\cdot)$ can generate task embeddings that can generalize to novel tasks, we conduct leave-one-out cross validation with all tasks in our task-model bank. Concretely, for each task considered as a novel task $T^{(n)}$, we use the rest of the tasks, along with their distance metric $d_g(\cdot, \cdot)$ estimated by GraphGym's exact but computationally expensive metric space, to train the projection function $g(\cdot)$. We calculate Kendall rank correlation over task similarities for Task Feature (without $g(\cdot)$) and Task Embedding (with $g(\cdot)$) against the exact task similarities. The average rank correlation and the standard deviation over ten runs is shown on Figure 4 (c). We find that with the proposed $g(\cdot)$, our task embeddings indeed correlate better with the exact task similarities, and therefore, generalize better to novel tasks.

**Ablation study on alternative task space design.** To demonstrate the superiority of the proposed task embedding, we further compare it with alternative task features. Following prior work (Yang et al., 2019), we use the normalized losses over the first 10 steps as the task feature. The results on OGB-Arxiv are shown in Table 2. Compared to AUTOTRANSFER's task embedding, the task feature induced by normalized losses has a lower ranking correlation with the exact metric and yields worse performance. Table 2 further justifies the efficacy of using the Kendall rank correlation as the metric for task embedding quality, as higher Kendall rank correlation leads to better performance.

Table 2: Ablation study on the alternative task space design versus AUTOTRANSFER's task embedding. We report the average test accuracy and the standard deviation OGB-Arxiv over ten runs.

|  | Kendall rank correlation | Test accuracy |
|---|---|---|
| Alternative: Normalized Loss | -0.07±0.43 | 68.13±1.27 |
| AUTOTRANSFER's Task Feature | 0.18±0.30 | 70.67±0.52 |
| AUTOTRANSFER's Task Embedding | 0.43±0.22 | 71.42±0.39 |

## 6 CONCLUSION

In this paper, we study how to improve AutoML search efficiency by transferring existing architectural design knowledge to novel tasks of interest. We introduce a *task-model bank* that captures the performance over a diverse set of GNN architectures and tasks. We also introduce a computationally efficient *task embedding* that can accurately measure the similarity between different tasks. We release GNN-BANK-101, a large-scale database that records detailed GNN training information of 120,000 task-model combinations. We hope this work can facilitate and inspire future research in efficient AutoML to make deep learning more accessible to a general audience.

ACKNOWLEDGEMENTS

We thank Xiang Lisa Li, Hongyu Ren, Yingxin Wu for discussions and for providing feedback on our manuscript. We also gratefully acknowledge the support of DARPA under Nos. HR00112190039 (TAMI), N660011924033 (MCS); ARO under Nos. W911NF-16-1-0342 (MURI), W911NF-16-1-0171 (DURIP); NSF under Nos. OAC-1835598 (CINES), OAC-1934578 (HDR), CCF-1918940 (Expeditions), NIH under No. 3U54HG010426-04S1 (HuBMAP), Stanford Data Science Initiative, Wu Tsai Neurosciences Institute, Amazon, Docomo, GSK, Hitachi, Intel, JPMorgan Chase, Juniper Networks, KDDI, NEC, and Toshiba. The content is solely the responsibility of the authors and does not necessarily represent the official views of the funding entities.

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

## A    ADDITIONAL IMPLEMENTATION DETAILS

**Runtime analysis.** We have empirically shown that AUTOTRANSFER can significantly improve search efficiency by reducing the number of trials needed to achieve reasonably good accuracy. The only overhead we introduced is the procedure of estimating task embeddings. Since we use a randomly initialized architecture, extracting each task feature only requires at most one forward and a few backward passes from a single minibatch of data. The wall-clock time depends on the size of the network and data structure. In our experiments, it typically takes a few seconds on an NVIDIA T4 GPU. We repeat the task feature extraction process 5 times for each anchor model, and for a total of 12 anchor models. Thus, the wall-clock time of the overhead for computing task embedding of novel task is within a few minutes. We note the length of this process is generally comparable to one trial of training on a small-sized dataset, and the time saved is much more significant for large-scale datasets.

**Details for task-model-bank training.** Our GNN model specifications are summarized in Table 3. Our code base was developed based on GraphGym (You et al., 2020). For all the training trials, We use the Adam optimizer and cosine learning rate scheduler (annealed to 0, no restarting). We use L2 regularization with a weight decay of 5e-4. We record losses and accuracies for training, validation and test splits every 20 epochs.

**Training details for AUTOTRANSFER.** We summarize the training procedure for the projection function $g(\cdot)$ in Algorithm 2. We set $U = 12$ and $R = 5$ throughout the paper. We use the same set of anchor model designs as those in GraphGym. We use a two-layer MLP with a hidden dimension of 16 to parameterize the projection function $g(\cdot)$. We use the Adam optimizer with a learning rate of 5e-3. We use margin $= 0.1$ and train the network for 1000 iterations with a batch size of 128. We adopt $K = 16$ when selecting the top K trials that are used to summarize design distributions.

**Details for adapting TPE, evolution algorithms.** We hereby illustrate how to compose the search algorithms with the transferred design priors. TPE is a Bayesian hyperparameter optimization method, meaning that it is initialized with a prior distribution to model the search space and updates the prior as it evaluates hyperparemeter configurations and records their performance. We replace this prior distribution with the task-informed design priors. As evolutionary algorithms generally initialize a large population and iteratively prune and mutate existing networks, we replace the random network initialization with the task-informed design priors. As we mainly focus on the few trial search regime, we set the fully random warm-up trials to 5 for both TPE and evolution algorithms.

Table 3: Design choices in our search space

| Type | Choices |
|---|---|
| Convolution | GeneralConv, GCNConv,SAGEConv,GINConv,GATConv |
| Number of heads | 1, 2, 4 |
| Aggregation | Sum, Mean-Pooling, Max-Pooling |
| Activation | ReLU, pReLU, leaky_ReLU, ELU |
| Hidden dimension | 64, 256 |
| Layer connectivity | Stack, Skip-Sum, Skip-Concat |
| Pre-process layers | 1, 2 |
| Message passing layers | 2,4,6,8 |
| Post-process layers | 2, 3 |
| Learning rate | 0.1, 0.001 |
| Training epochs | 200, 800, 1600 |

## B    ADDITIONAL DISCUSSION

**Limitations.** In principle, AUTOTRANSFER leverages the correlation between model performance rankings among tasks to efficiently construct model priors. Thus, it is less effective if the novel task has large task distances with respect to all tasks in the task-model bank. In practice, users can continuously add additional search trials to the bank. As the bank size grows, it will be less likely that a novel task has a low correlation with all tasks in the bank.

---

**Algorithm 2** Training Pipeline for the projection function $g(\cdot)$

---

**Require:** Task features $\{z_f^{(i)}|T^{(i)}\}$ extracted for each task from the task-model bank. Distance measure $d_g(\cdot, \cdot)$ estimated in GraphGym.

1: **for** each iteration **do**
2:     Sample $T^{(i)}, T^{(j)}, T^{(k)}$
3:     $z_e^{(i)}, z_e^{(j)}, z_e^{(k)} \leftarrow g(z_f^{(i)}), g(z_f^{(j)}), g(z_f^{(k)})$
4:     $y \leftarrow 1$ if $d_g(T^{(i)}, T^{(j)}) < d_g(T^{(i)}, T^{(k)})$ else $-1$
5:     Optimize objective function $\mathcal{L}_r(z_e^{(i)}, z_e^{(j)}, z_e^{(k)}, y)$ in Eq. 2
6: **end for**

---

**Social Impact.** Our long-term goal is to provide a seamless GNN infrastructure that simplifies the training and deployment of ML models on structured data. Efficient and robust AutoML algorithms are crucial to making deep learning more accessible to people who are interested but lack deep learning expertise as well as those who lack the massive computational budget AutoML traditionally requires. We believe this paper is an important step to provide AI tools to a broader population and thus allow for AI to help enhance human productivity. The datasets we used for experiments are among the most widely-used benchmarks, which should not contain any undesirable bias. Besides, training/test losses and accuracies are highly summarized statistics that we believe should not incur potential privacy issues.

## C   ADDITIONAL RESULTS

**Search efficiency.** We summarize the average number of trials needed to surpass the average best accuracy found by TPE with 30 trials in Table 4. We show that AUTOTRANSFER reduces the number of explored architectures by *an order of magnitude*.

Table 4: Average number of search trials needed to surpass the average best result found by TPE with 30 trials

|  | Node | | | Graph | | |
|---|---|---|---|---|---|---|
|  | Physics | CoraFull | OGB-Arxiv | COX2 | IMDB | PROTEINS |
| Num. of Trials | 3 | 2 | 3 | 4 | 3 | 6 |
| Accuracy | 96.64±0.42 | 67.85±1.31 | 71.42±0.39 | 82.96 ± 1.75 | 52.33±2.13 | 80.21±1.21 |

**Ablation study on number of anchor models and task embedding design.** We empirically demonstrate how the number of anchor models affect the rank correlation in Table 5. While 3 anchor models are not enough to capture the task distance, we found that 9 and 12 have a satisfactory trade-off between capturing task distance and computational efficiency. Furthermore, we empirically in Table 6 demonstrate that the learned task embedding space is superior to the proposed task feature space in terms of correlation as well as final search performance.

Table 5: Average Kendall rank correlation of similarity rankings of the other tasks with respect to the central task between the proposed method and GraphGym.

| Num. of anchor models | 3 | 6 | 9 | 12 |
|---|---|---|---|---|
| Task Feature | 0.03±0.34 | 0.11±0.36 | 0.16±0.34 | 0.18±0.30 |
| Task Embedding | 0.12±0.28 | 0.26±0.30 | 0.36±0.24 | 0.43±0.22 |

**Visualizing model designs.** We visualized the transferred design distributions and ground truth design distributions on the TU-PROTEINS dataset in Figure 5, as well as Coauthor-Physics dataset in Figure 6. We could observe that the transferred design distributions have a positive correlation on most of the design choices.

Table 6: Ablation study of the number of anchor models as well as task embedding vs. task feature on OGB-Arxiv with 3 trials. We report the average test accuracy and the standard deviation over ten runs.

| Num. of anchor models | 3 | 6 | 9 | 12 |
|---|---|---|---|---|
| Task Feature | 69.42±0.82 | 69.86±0.78 | 70.41± 0.59 | 70.67±0.52 |
| Task Embedding | 69.80± 0.75 | 70.59±0.63 | 71.16±0.47 | 71.42±0.39 |

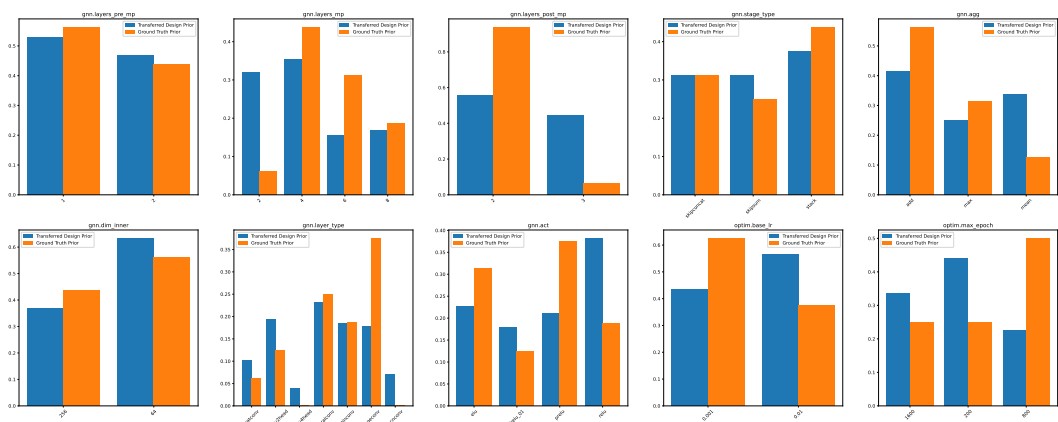

Figure 5: We plot the transferred design distributions and ground truth design distributions on the TU-PROTEINS dataset.

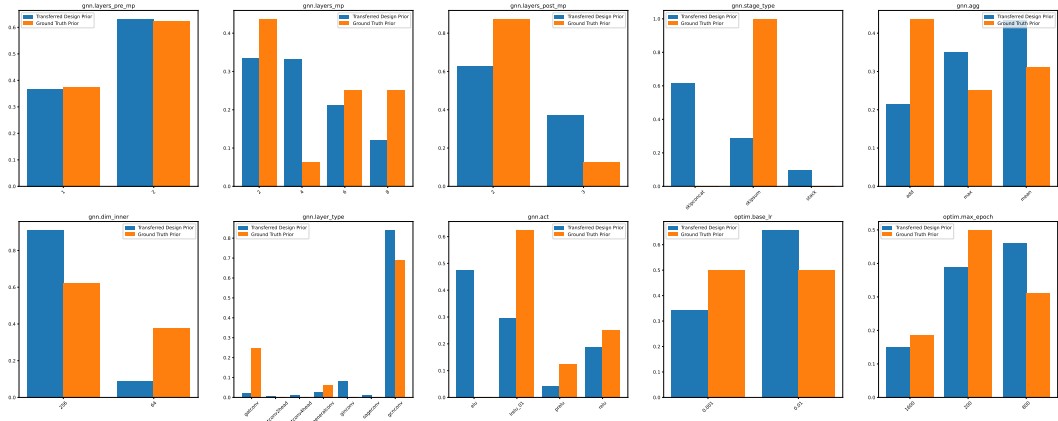

Figure 6: We plot the transferred design distributions and ground truth design distributions on the Coauthor-Physics dataset.

