# OpenReview forum: "AutoTransfer: AutoML with Knowledge Transfer - An Application to Graph Neural Networks"
_ICLR.cc/2023/Conference — ICLR 2023 poster_

### Official Review · Reviewer_SdtT · 2022-10-24

**Confidence:** 3
**Correctness:** 4
**Technical Novelty And Significance:** 4
**Empirical Novelty And Significance:** 3
**Recommendation:** 8

**Clarity, Quality, Novelty And Reproducibility:**

- There is already another transfer learning approach named "Auto-Transfer" and frankly speaking the name is not very innovative or self-describing: Murugesan, Keerthiram, et al. "Auto-Transfer: Learning to Route Transferrable Representations." arXiv preprint arXiv:2202.01011 (2022).
- Why is only Kendall's correlation coefficient considered? There could be different degrees of similarity, something like weak similarity where only a top-k ranking needs to be consistent. However, since one is not interested in the precise ranking of low performers, one could argue that this is less relevant for comparing similar datasets. Maybe a correlation measure such as normalized discounted cumulative gain (NDCG) would be a better choice here? The authors state that they see a justification for Kendall's tau in the results but do not discuss other ranking correlation measures.

### Clarity
The clarity is overall quite good. Language is clear and the paper is well structured.

### Quality and Novelty
The quality of the paper is very good and the approach is novel - at least to the best of my knowledge.

### Reproducibility
Important details regarding the methods evaluated are missing. In particular how competitors have been parameterized, what kind of evolution was used as a baseline and how it was configured etc.

**Strength And Weaknesses:**

[+] The provided benchmark dataset is valuable and hopefully facilitates and enables research on transfer learning graph neural networks.

[+] The proposed transfer learning approach seems to perform reasonably well refining the state of the art.

[+] To the best of my knowledge, the proposed method is novel.



[-] The authors state that all GNNs can be described in terms of three stages: pre-processing with MLPs, message passing layers, and post-processing with MLPs. However, I would argue that this is quite restrictive and not able to reflect all possible architectures. See for example

Lee, Junhyun, Inyeop Lee, and Jaewoo Kang. "Self-attention graph pooling." International conference on machine learning. PMLR, 2019.

and

Ma, Yao, et al. "Graph convolutional networks with eigenpooling." Proceedings of the 25th ACM SIGKDD international conference on knowledge discovery & data mining. 2019.

where pooling operations are added between the convolutions resulting in iterative changes of the graph structure. Of course, such a benchmark database needs to be restricted in some sense so that the set of candidates is reasonably limited. However, I would suggest to rephrase the statement in Section 4.2 a little bit so that it makes clear that the benchmark limits itself to such GNNs.

[-] It is not clear why the authors chose the mentioned datasets for evaluation and why results on the other datasets are not reported.

**Summary Of The Paper:**

The paper "AutoTransfer: AutoML with Knowledge Transfer - An Application to Graph Neural Networks" makes two major contributions: First, the authors devise a benchmark for searching graph neural networks, consisting of 12 task and comprising 120,000 evaluated task-model combinations. Second, they propose a method for transfer learning, suggesting a prior based on past experience for searching neural architectures on a new dataset.

**Summary Of The Review:**

The paper makes good contributions to the field and provides some new artefacts for the community which can be quite useful to foster future research on NAS for GNNs. The paper has some minor issues which can however be fixed quite easily. Therefore, I would recommend to accept the paper under the condition that the issues are fixed for the camera ready version.

---

> ### Author Response · Authors · 2022-11-12
> **Response to Reviewer SdtT**
>
> We thank the reviewer for the valuable feedback and insightful comments. We also appreciate the reviewer for thinking this paper novel and valuable. We hope the newly provided clarification could help to further strengthen our work.
>
> *Q1. “Describing GNN into three stages is quite restrictive and not able to reflect all possible architectures. I would suggest to rephrase the statement in Section 4.2 a little bit so that it makes clear that the benchmark limits itself to such GNNs.”*
>
> We thank the reviewer for the suggestion. We have included additional references and rephrased statement with limitation clarified in the corresponding section.
>
> *Q2. “It is not clear why the authors chose the mentioned datasets for evaluation and why results on the other datasets are not reported.”*
>
> We thank the reviewer for the question. We have the performance results on all the datasets in the GNN-BANK-101 that we have prepared. Our consideration is mainly to report on datasets that are widely used by previous works as well, e.g., node classification datasets that we reported follows Table 3 in [1].
>
> *Q3. “There is already another transfer learning approach named "Auto-Transfer" and frankly speaking the name is not very innovative or self-describing.”*
>
> We appreciate the reviewer for the suggestion. Indeed, we could use a more self-describing name for the proposed method, such as “AutoTaskTransfer”. We will finalize on a better naming in the revised paper version.
>
> *Q4. “Why is only Kendall's correlation coefficient considered? The authors state that they see a justification for Kendall's tau in the results but do not discuss other ranking correlation measures.”*
>
> We thank the reviewer for the suggestion. We agree that other ranking correlation measures, such as normalized discounted cumulative gain (NDCG) can potentially be a good choice, which focus more about top performance models. Nevertheless, we believe using Kendall rank correlation is sufficient to demonstrate the effectiveness of the proposed method. We thank the reviewer for the constructive suggestion, and we believe it could be a valuable future work to explore the effect of ranking correlation measures on the quality of task transfer.
>
> [1] Qin, Yijian, et al. "Graph differentiable architecture search with structure learning." Advances in Neural Information Processing Systems 34 (2021): 16860-16872.

---

### Official Review · Reviewer_1hga · 2022-10-25

**Confidence:** 4
**Correctness:** 3
**Technical Novelty And Significance:** 3
**Empirical Novelty And Significance:** 3
**Recommendation:** 6

**Clarity, Quality, Novelty And Reproducibility:**

The paper is well-written and easy to follow. The proposed methods are interesting.

**Strength And Weaknesses:**

### Strengths
1. The paper introduces an interesting method for transferring knowledge from a previously searched task to a new coming task with a task-model bank design.
2. It proposes an interesting task embedding with several random-initialized anchor models, and the experiments show that the task similarity is promising.
3. The methods are tested on multiple benchmarks and are proven to be efficient and performant.
4. It releases a large task-model dataset that can inspire future research.

### Weaknesses
1. How are the anchor models selected? I failed to find the architecture design of the anchor models, and it would be good to see how diverse the anchor models should be.
2. The graph architectures used in the paper are quite limited. For example, on the OGB leaderboard, many different GNN (non-GNN) algorithms are applied and achieve significantly better results. It would be interesting to have some of those models in the search space.
3. It would be interesting to see some examples of the models selected for a group of tasks, to get some insights of the model design.


**Summary Of The Paper:**

The paper proposes a novel AutoML method, AutoTransfer, for graph learning tasks. It introduces a task-model bank that can transfer the model design to similar new tasks. The experiments indicate that the method is efficient and improves the performance of the found models.

**Summary Of The Review:**

The paper proposes an interesting method for transferring knowledge in AutoML for graph learning tasks.

---

> ### Author Response · Authors · 2022-11-12
> **Response to Reviewer 1hga**
>
> We thank the reviewer for the valuable feedback and insightful comments. We also appreciate the reviewer for thinking this method interesting, and the paper is well written and easy to follow. We hope the newly provided clarification could help to further strengthen our work. We would appreciate the reviewer to increase the score if these clarifications could resolve the raised concerns.
>
> *Q1. “How are the anchor models selected?”*
>
> We thank the reviewer for the question. We have mentioned in the Appendix that we followed the selection process introduced in GraphGym [1], and further conducted an ablation study on the selection of anchor models. We apologize for not making it clear in the main paper due to the page constraint.
> Specifically, we first sample D random GNN designs from the design space. Then, we apply these designs to a fixed set of GNN tasks, and record each GNN’s average performance across the tasks. The D designs are ranked and evenly sliced into M groups, where models with median performance in each group are selected.
>
> *Q2. “The graph architectures used in the paper are quite limited. For example, on the OGB leaderboard, many different GNN (non-GNN) algorithms are applied and achieve significantly better results.”*
>
> We thank the reviewer for the comment. Since the focus of this paper is to propose a novel AutoML search algorithm rather than pursuing the best benchmark results, we carefully created a general GNN search space that work across different graph learning tasks, and rigorously compared SOTA AutoML algorithms **under the same search space**. We are aware of other GNN (non-GNN) methods, but they cannot be summarized into a similar general search space as message passing GNNs. For example, methods like Correct and Smooth (C&S) achieved SOTA performance on node-levels tasks, but they are not applicable to graph-level tasks.
> Nevertheless, since the proposed AutoTransfer is a general AutoML algorithm, it can be readily extended to non-GNN algorithms once a standard search space for non-GNN algorithms have been prepared. We thank for the reviewer for the insightful suggestion, and we believe it could be a valuable future work to explore.
>
> *Q3. “It would be interesting to see some examples of the models selected for a group of tasks, to get some insights of the model design.”*
>
> We thank the reviewer for the comment. We included additional visualization results in Appendix C. Concretely, we visualized the transferred design distributions and ground truth design distributions of top performance models. We could observe that the transferred design distributions have a positive correlation on most of the design choices. We apologize for not being able to include such examples in the main paper due to the page constraint.
>
> [1] Jiaxuan You, Rex Ying, Jure Leskovec. "Design Space for Graph Neural Networks." NeurIPS 2020

---

### Official Review · Reviewer_uQiP · 2022-10-31

**Confidence:** 3
**Correctness:** 3
**Technical Novelty And Significance:** 3
**Empirical Novelty And Significance:** 3
**Recommendation:** 6

**Clarity, Quality, Novelty And Reproducibility:**

Clarity: The paper is easy-to-understand.
Quality: Overall, the paper is technically sound. Weaknesses pointed out in the previous section.
Novelty: To the best of the reviewer's knowledge, the algorithm of obtaining task embedding is novel
Reproducibility: The reviewer has not checked reproducibility but the author has provided enough details in the appendix.

**Strength And Weaknesses:**

Strength

1. The idea of utilizing FIM to produce task-feature and further finetune it with ranking loss is novel.
2. Ablation analysis justifies the effectiveness of the task embedding algorithm proposed by the paper.


Weaknesses

1. It seems that "[AAAI2021] One-shot graph neural architecture search with dynamic search space" also discussed task transfer. The author may need to compare with one-shot NAS or clarify the difference.
2. The author pointed out that there are three stages in GNN architectures: 1) pre-processing, 2) message-passing, 3) post-processing. The author only considered message-passing layers and pointed out that it is commonly regarded as the most important part of GNN. However, since different tasks adopt different pre-/post-processing modules, it is not convincing to directly state tat message-passing is the most important.
3. It will be good if the author can list the performance of SOTA algorithm in each dataset discussed in Table 1. This will help readers understand the gap between NAS models and human-engineered SOTA architectures.


**Summary Of The Paper:**

The paper studied NAS algorithm for GNN. The author proposed to transfer the prior architectural design knowledge to the novel task of interest. The author first extracts scale-invariant measures based on Fisher Information Matrix (FIM). The scale-invariant measures are used to form the task feature, which are further projected to task embeddings. The task embeddings are learned via a ranking loss built on top of GraphGym's extact metric space. Experiments justified the importance of the task embeddings in the AutoTransferer algorithm. The AutoTransferer also outperforms baselines like GraphNAS,


**Summary Of The Review:**

Voted for weak accept because the algorithm is novel. However, I still have concerns pointed out in the Weakness section.

---

> ### Author Response · Authors · 2022-11-12
> **Response to Reviewer uQiP**
>
> We thank the reviewer for the valuable feedback and insightful comments. We also appreciate the reviewer for finding our proposed algorithm novel. We hope the newly provided clarification could help to further strengthen our work. We would appreciate the reviewer to increase the score if these clarifications could resolve the raised concerns.
>
> *Q1. “It seems that "[AAAI2021] One-shot graph neural architecture search with dynamic search space" also discussed task transfer. The author may need to compare with one-shot NAS or clarify the difference.”*
>
> We thank the reviewer for the suggestion. We were aware of the suggested paper [1] and have already included it  in the reference. Meanwhile, we have compared with two one-shot NAS methods, DARTS and GASSO, in the Table 1. The task transfer discussed in [1] is about doing architecture search on a small subset of the whole dataset and later transfer the best searched architecture to the full dataset. We wish to point out that this type of acceleration is orthogonal and complementary to our setting, i.e., after getting the transferred design prior, we can also search on a small subset instead of the whole dataset. We thank for the reviewer for the insightful suggestion, and we believe it could be a valuable future work to explore.
>
> *Q2. “The author only considered message-passing layers and pointed out that it is commonly regarded as the most important part of GNN. However, since different tasks adopt different pre-/post-processing modules, it is not convincing to directly state that message-passing is the most important.”*
>
> We thank the reviewer for the comment. We fully agree that other modules in our search space, such as pre-/post-processing modules, are important as well. When computing FIM, although we only explicitly consider Jacobians with respect to weights of the message passing layers, **weights of pre-/post-processing modules layers are also implicitly coupled in the forward pass. Furthermore, due to the different input data and prediction heads, Jacobians of pre/post-processing layers are not directly comparable due to the size mismatch.** We have modified the paper to clarify this point.
>
> *Q3. “It will be good if the author can list the performance of SOTA algorithm in each dataset discussed in Table 1.”*
>
> We thank the reviewer for the suggestion. Since the focus of this paper is to propose a novel AutoML search algorithm rather than pursuing the best benchmark results, we rigorously compared different SOTA AutoML algorithms **under the same search space**. On the other hand, existing SOTA performance on datasets in Table 1 are obtained without a pre-defined search space, and usually involves additional post-processing steps like Correct and Smooth (C&S) and model ensemble, giving these methods additional advantages over the existing methods in Table 1. To avoid confusing the readers about the key message in Table 1, we did not include such results in the table. To help readers better understand the results, we have included results of popular GNN architectures (e.g., GCN, GAT) in Table 1.
>
> [1] Li, Yanxi, et al. "One-shot graph neural architecture search with dynamic search space." Proceedings of the AAAI Conference on Artificial Intelligence. Vol. 35. No. 10. 2021.

---

### Decision · Program_Chairs · 2023-01-20

**Decision:**

Accept: poster

**Justification For Why Not Higher Score:**

The contribution of this paper is strictly limited to the application of existing ideas to the graph domain due to the following reasons:
- The idea of retrieving architectures from a large model zoo is not novel, since it has been already done in [Jeong et al. 21].
- The idea of using Fisher information matrix for neural architecture search was proposed in [Le et al. 22].

**Justification For Why Not Lower Score:**

The proposed framework as well as the large-scale GNN database can serve as good benchmark and baselines for future research on NAS for GNNs.

**Metareview: Summary, Strengths And Weaknesses:**

This paper proposes a neural architecture search framework for GNNs, which retrieves the most similar GNN architectures from a large database of randomly initialized GNNs based on their task-level similariies. The task-level similarities of the GNNs are computed with a modified version of Fisher Information Matrix embedding [Archille et al. 19] to deal with the irregularities in the graph data. The experimental results verify the effectiveness of the proposed framework.

All reviewers leaned toward acceptance, as they considered the idea of retrieving neural architectures based on the task similarity encoded using Fisher Information Matrix as novel, the construction of the large database of GNNs for transferable NAS as useful for future research, and the proposed framework to be practically effective. However, the reviewers also had concerns regarding not considering other aspects that can have large impact on the GNN models (e.g. pooling), comparison against state-of-the-art NAS approaches, and lack of discussion of closely related works.

Despite the overall positive sentiment among the reviewers, I am concerned about the lack of novelty in the general AutoML perspective, and the overclaimed contributions of the work, which I believe is due to insufficient literature study.

- The idea of retrieving an architecture from a model zoo based on the task similarity is not novel, since [Jeong et al. 21] proposed the same idea, with a different task encoding. The authors argue that [Jeong et al. 21] cannot distinguish between two tasks, but the embedding simply has less discriminative power and the high-level ideas are essentially the same.

- The idea of using Fisher Information Matrix to encode task similarity is the contribution of [Archille et al. 19], and using Fisher Information Matrix for NAS has been already done in [Le et al. 22].

- However, the authors did not discuss these works, which are either conceptually similar or essentially based on the same method for a non-graph domain. The title “AutoTransfer: AutoML with Knowledge Transfer - An Application to Graph Neural Networks” seems to suggest the method is applicable to non-graph domains, but there is no concrete formulation or experimental results on other domains.

I believe that the contribution of the proposed work is strictly limited to the extension of them to the graph domain.
Construction of a large-scale GNN database
Modification of the FIM-based task embedding for encoding graphs.

Yet, the title of the paper, the name of the proposed framework, and the claimed contributions of the authors do not accurately convey what are actually done. In the final revision of the paper, the authors should clearly acknowledge that similar ideas have been proposed in non-graph domains and revise the title, naming, and the claimed contributions accordingly.

[Archille et al. 19] Task2Vec: Task Embedding for Meta-Learning, ICCV 2019.
[Jeong et al. 21] Task-Adaptive Neural Network Search with Meta-Contrastive Learning, NeurIPS 2021
[Le et al. 22] Fisher Task Distance and Its Application in Neural Architecture Search, IEEE 2022


**Note From Pc:**

if the above contains the word "oral" or "spotlight" please see: "oral" presentation means -> notable-top-5% and "spotlight" means -> notable-top-25%. As stated in our emails, we are disassociating presentation type from AC recommendations